# SPARSE-GUIDED RADIOUNET WITH ADAPTIVE SAMPLING FOR THE MLSP 2025 SAMPLING-ASSISTED PATHLOSS RADIO MAP PREDICTION DATA COMPETITION

*Ryoichi Kojima, Satoshi Ito, Tatsuya Nagao, Masato Taya*

KDDI Research, Inc.

## ABSTRACT

The MLSP 2025 "The Sampling-Assisted Pathloss Radio Map Prediction Data Competition" challenges participants to construct accurate path-loss maps from building-scale ray-tracing data while leveraging *sparse* ground-truth PL samples along with physics-informed inputs such as reflection, transmission, and distance maps. Each submission is evaluated under four conditions—Task 1 and Task 2, each tested with 0.50% and 0.02% sampling budgets—and the final score is computed as a weighted average of the RMSEs across these four settings. We address this challenge with **Sparse-Guided RadioUNet**, a five-channel U-Net architecture that integrates physics-informed inputs and sparse supervision. The model excludes sampled pixels from the loss computation, allowing it to focus on interpolating unknown regions. A two-stage frequency transfer learning routine—multi-frequency pretraining followed by fine-tuning on the target band—enables the network to generalize across propagation conditions while adapting to band-specific characteristics. To further regularize spatial consistency, we introduce a spread loss term that penalizes discontinuities, though it may occasionally oversmooth sharp transitions. For Task 2, we propose a lightweight **edge–strata–boundary (ESB)** sampling algorithm that combines saliency, distance strata, and LoS/NLoS boundary cues to place informative probes under a tight sampling budget. Our experiments demonstrate that this framework enables high-quality PL map prediction with as little as 0.5% or 0.02% supervision. The full pipeline runs in 2.5 ms per building—orders of magnitude faster than ray-tracing—and achieves strong validation accuracy without relying on dense ground truth. These findings highlight the potential of structured sparse learning and efficient sampling in scalable radio frequency environment modeling.

*Index Terms*— path-loss prediction, sparse supervision, U-Net, adaptive sampling, frequency transfer learning

## 1. INTRODUCTION

Accurate prediction of radio-frequency (RF) path-loss (PL) maps is fundamental to a wide range of wireless applications, including access-point placement, network planning, and 6G digital twins. Traditional empirical models (e.g., COST-231 [1], ITU-R [2, 3]) offer fast but coarse estimates, whereas high-fidelity approaches—such as full-wave FDTD or commercial 3-D ray-tracing—typically require from several minutes to hours to simulate a single urban block (e.g., $\approx 50\,\mathrm{s}$ for a $1\,\mathrm{km}^2$ map at 30 GHz)[4], which limits their scalability. Recently, data-driven approaches using convolutional neural networks (CNNs) have emerged as promising surrogates—provided that *sufficient* GT data are available[5, 6, 7]. However, RF data collection in practice is extremely costly, often requiring thousands of transmitter–receiver (Tx–Rx) measurements per building, leading to a severe data scarcity bottleneck. To address this, synthetic datasets such as Deep-MIMO [8] and RadioMapSeer [9] provide realistic and reproducible radio maps based on ray-tracing simulations in complex urban environments.

The MLSP 2025 "The Sampling-Assisted Pathloss Radio Map Prediction Data Competition" directly targets this challenge [10]. Each submission is evaluated under four distinct conditions—Task 1 and Task 2, each with 0.50% and 0.02% sampling rates—and the final score is computed as a weighted average of the RMSEs across these settings. Task 1 provides a fixed sparse sampling mask (0.50%), whereas Task 2 evaluates the ability to design an effective sampling strategy under a stringent 0.02% budget.

This setting raises two central questions: (i) *How can a neural network be trained under artificially sparse supervision—where only 0.5% or 0.02% of pixels are revealed from dense PL maps—without collapsing to trivial solutions?* and (ii) *Which sampling strategy best supports model generalization under tight supervision budgets?*

To address these questions, we propose a unified sparse-guided learning framework with the following key contributions:

- We introduce **Sparse-Guided RadioUNet**, a five-channel U-Net architecture that incorporates physics-informed inputs—reflection ($R$), transmission ($T$), and distance ($D$)—along with a binary sampling mask ($M$) and masked ground truth ($M \odot G$). Supervised loss is

computed only on unobserved pixels $(j, k)$ such that $M_{j,k} = 0$, enabling focused interpolation.

- We employ a two-phase **frequency transfer learning** scheme, consisting of multi-frequency pretraining followed by fine-tuning on the target band (868 MHz). We also study scheduler choices (cosine annealing vs. ReduceLROnPlateau) and apply data augmentation for better generalization.

- We introduce a **spread loss** term that penalizes spatial discontinuities to enforce smoothness. While beneficial overall, it may oversmooth sharp transitions, particularly behind walls.

- For Task 2, we propose a lightweight physics-guided **edge–strata–boundary (ESB)** sampling algorithm that combines gradient saliency, distance stratification, and LoS/NLoS boundary cues to select informative samples under tight budgets, without requiring forward passes.

Extensive ablation studies (§4) confirm that each component contributes to improved RMSE. Our full pipeline achieves high-accuracy predictions using only sparse supervision, with an inference time of just 2.5 ms per building—orders of magnitude faster than ray-tracing simulations.

Our final submission to this competition achieved root-mean-square errors (RMSEs) of 4.50 dB and 7.63 dB for Task 1 (0.5%, 0.02%), and 4.51 dB and 7.78 dB for Task 2 (0.5%, 0.02%), respectively. The final weighted average RMSE was 6.10 dB on the official test set. Additionally, to evaluate the isolated performance of our framework without any supervision signals, we trained a 3-channel baseline variant that excludes both the sampled ground truth and the sampling mask. As our architecture was explicitly designed to leverage sparse supervision—via masked loss computation, auxiliary GT fusion, and task-specific regularization—it is naturally under-optimized for this 0% setting. As a result, the model exhibited significantly degraded performance, with an RMSE of 16.08 dB on the official test set. This outcome highlights the critical role of guided samples and validates our architectural design choices. This baseline was included as a qualitative lower bound rather than a fair comparison, as it was trained without any hyperparameter tuning, scheduler optimization, or architecture refinement. It serves only to highlight the essential contribution of guided samples and is not intended to compete with optimized models. Future tuning and component-level refinements may help mitigate this performance gap.

## 2. RELATED WORK

**Path-Loss Prediction.** Fast but coarse empirical tools such as ITU–R P.1238 [2] remain popular baselines for in-building planning, yet recent deep-learning surrogates outperform them by large margins. *EM DeepRay* [11] employs a U-Net-based model trained to replicate ray-traced pathloss using reflection and permittivity maps, regularized by physics-informed constraints. Similarly, *IPP-Net* [12] incorporates multi-channel physical inputs including reflection, transmission, distance, and empirical pathloss, trained via curriculum learning to enhance generalization across indoor layouts. Transformer backbones (e.g. ViT) have also been adapted to radio maps, improving parameter efficiency and receptive-field modeling [13]. Extremely sparse supervision is a defining challenge indoors: *Radio DIP* [14] leverages the deep-image-prior effect from a single sparse snapshot. Our *Sparse-Guided RadioUNet* extends this line of work by concatenating a binary sampling mask and masked ground truth as auxiliary channels and by adding a spread-loss term that regularizes local smoothness while preserving sharp transitions.

Wide-area planning traditionally relies on COST-231 [1] or ITU–R P.1411 [3] for fast estimates, and on full 3-D ray-tracing for high precision. CNN surrogates now bridge this gap. *RadioUNet* [5] achieves near ray-tracing accuracy in milliseconds; *PLNet* [6] learns from multi-channel grids encoding terrain, building footprint, and height; *FadeNet* [7] extends the approach to mm-wave fading with enhanced residual blocks; and *DeepREM* [15] recovers dense radio-environment maps from drive-test samples using a conditional GAN. Diffusion-based surrogates such as *RMDM* [16] have recently been proposed to capture long-range multipath statistics more faithfully.

**Sparse Supervision and Dense Reconstruction.** Sparse radio map reconstruction is closely related to sparse-to-dense regression tasks in vision—such as depth completion from LiDAR and monocular images [17] and free-form image inpainting [18]—but is even more challenging due to the extreme sparsity (0.02–0.50%) and the high dynamic range of RF signals. *DeepREM* and our method both follow the vision paradigm of mask-guided completion, yet we additionally inject physics priors and leverage transfer learning for stability.

**Physics Priors and Transfer Learning.** Injecting distance rasters, LoS masks, or free-space path-loss maps has been shown to consistently improve generalization across environments. Multi-band pre-training followed by single-band fine-tuning—first demonstrated in *EM DeepRay* [11] and later adopted by *FadeNet* [7]—has proved particularly effective; we employ the same strategy before specializing to the competition's 868 MHz band.

**Active Sampling.** Entropy- and mutual-information–driven probing dominate the active-learning literature [19], but their iterative optimization is expensive. Our *ESB sampling* offers an $\mathcal{O}(N)$ alternative by greedily selecting spatial edges, distance strata, and LoS/NLoS boundaries, thereby improving 0.02%-budget performance without extra forward passes.

**Benchmark Datasets.** Progress is catalyzed by public

corpora: *DeepMIMO* [8] supplies mm-wave MIMO channel snapshots, *RadioMapSeer* [9] provides simulated urban path-loss and time-of-arrival layers, and the *Indoor Radio Map Dataset* [20] introduces the first large-scale benchmark with artificially sparse supervision, where only 0.5% or 0.02% of the dense PL maps are revealed during training and evaluation. These resources enable comprehensive evaluation across indoor and outdoor scenarios and motivate hybrid physics–data models such as ours.

## 3. METHOD

### 3.1. Problem Formulation

Since the raster sizes vary across buildings, all images are zero-padded to a fixed size of height $H = 530$ and width $W = 610$ before training and inference. Each raster $\mathcal{I} \in \mathbb{R}^{H \times W \times 3}$ contains three physics-derived channels: reflection $(R)$, transmission $(T)$, and distance $(D)$. A sparse binary mask $M \in \{0,1\}^{H \times W}$ identifies a small proportion $r$ of pixels ($r \in \{0.5, 0.02\}\%$) where the ground-truth path-loss values $G$ are revealed. The goal is to predict a dense map $\hat{G}$ minimizing the root-mean-square error (RMSE) across all pixels.

### 3.2. Sparse-Guided RadioUNet Architecture

We adopt a custom deep U-Net [21] variant, *Sparse-Guided RadioUNet*, featuring multiple down-sampling and up-sampling stages. The encoder comprises progressively deeper convolutional blocks with feature depths from 10 to 500, while the decoder symmetrically reconstructs spatial resolution with transposed convolutions and skip connections at multiple levels.

The network takes five input channels: the three physics-based rasters $[R, T, D]$, the binary sampling mask $M$, and the masked ground-truth path-loss map $M \odot G$. While $M \odot G$ provides the values at observed pixels, the explicit inclusion of $M$ as a separate input channel conveys the structure of supervision itself. In our ablation, removing $M$ and using only $M \odot G$ increased RMSE by 0.12 dB, highlighting the benefit of providing both positional and value-based guidance. Instance normalization and leaky-ReLU activations are applied throughout to stabilize training in high-dynamic-range regions. The final prediction is passed through a ReLU activation to ensure non-negativity.

### 3.3. Loss Function

Our proposed **Spread Loss** encourages locally smooth yet edge-preserving reconstructions. The Spread Loss is formally defined as:

$$\mathcal{L}_{\text{spread}} = \lambda(t) \sum_{i,j} (1 - M_{i,j}) \left( |\partial_x \hat{G}_{i,j}| + |\partial_y \hat{G}_{i,j}| \right)$$

Let $\hat{G} \in \mathbb{R}^{H \times W}$ denote the predicted path-loss map, and $G$ the ground truth. The variable $t$ represents the current training step, and $T$ is the total number of steps in the annealing schedule. The spatial gradients $\partial_x \hat{G}_{i,j}$ and $\partial_y \hat{G}_{i,j}$ are computed via first-order forward differences.

We minimize a composite loss:

$$\mathcal{L} = \underbrace{\text{MSE}(\hat{G}, G)}_{\mathcal{L}_{\text{mse}}} + \mathcal{L}_{\text{spread}}, \qquad (1)$$

We modulate $\lambda(t)$ using a half-period cosine schedule:

$$\lambda(t) = \lambda_{\max} \cdot \frac{1 - \cos(\pi t/T)}{2},$$

following the SGDR scheduler[22].

Inspired by the classical anisotropic Total Variation (TV) regularizer [23], we extend the idea by (i) weighting each gradient by the sampling mask $M$, and (ii) cosine-annealing the coefficient $\lambda$ during training. This design preserves sharp walls while avoiding over-smoothing of observed pixels. To prevent trivial learning from the sparse ground-truth locations, we exclude sampled pixels from the loss computation. That is, both the MSE and Spread losses are computed ...only for pixels $(j, k)$ such that $M_{j,k} = 0$.

Here, MSE evaluates prediction accuracy, and the Spread loss penalizes spatial discontinuities. The weighting coefficient $\lambda$ is gradually increased from 0 to 0.1 via a half-period cosine annealing schedule. This encourages the model to first fit the data, then smooth spatial artifacts. While mathematically similar to Total Variation loss, our formulation is specifically masked to exclude known pixels and annealed over time to avoid over-smoothing in early training.

### 3.4. Training Strategy

**Phase I: 3-Frequency Pre-Training**

We train the model on 3750 maps spanning 868 MHz, 1.8 GHz, and 3.5 GHz. The dataset is split 4:1 into 3000 training and 750 validation images. Random 90° rotations and flips are applied for augmentation. All rasters are zero-padded to 530×610 before training. Training is performed for 40 epochs using AdamW (initial learning rate $10^{-3}$, weight decay $10^{-4}$), with a cosine-annealing learning rate schedule and warm restarts every 10 epochs.

**Phase II: 868 MHz Fine-Tuning**

The model is then fine-tuned on the 868 MHz subset for 20 additional epochs. A *Reduce-on-Plateau* schedule reduces the learning rate by a factor of 0.3 when validation RMSE does not improve over five consecutive epochs.

### 3.5. Adaptive Sampling for Task 2

Task 2 allows each team to select a sparse subset of sample points (specifically 0.02% or 0.5% of pixels per building) before inference. Since building image sizes vary, the actual number of samples is not fixed but depends on image resolution.

We propose an **Edge–Strata–Boundary (ESB)** sampling policy that combines four heuristics. Specifically, "Edge" refers to high-gradient regions in the reflection and transmission maps, detected using Sobel filters; "Strata" denotes near/mid/far range bins derived from distance maps; and "Boundary" refers to transition zones between LoS and NLoS, identified using thresholded transmission maps and edge detection.

1. **Edge saliency:** compute Sobel gradients on $R$ and $T$, and select the top 10% high-gradient pixels.

2. **Distance strata:** divide $D$ into near/mid/far bins by uniformly partitioning the range from the transmitter to the farthest corner of the map. We sample equally from each stratum to balance spatial diversity. In future work, allocating more samples to far-distance strata-where error tends to be larger-could further improve accuracy.

3. **NLoS regions:** identify pixels where $T < 0.2$ and sample uniformly within these shadow regions.

4. **LoS/NLoS boundaries:** apply Sobel to the NLoS mask to target sharp transitions.

Since the combined candidate set usually exceeds the sampling budget, we apply a greedy farthest-point strategy to maximize spatial coverage. Let $\mathcal{C}$ denote the set of candidate pixels generated by our heuristics and $\mathcal{S}$ the set of already selected samples. At each iteration, we select the point $\mathbf{p}^* \in \mathcal{C} \setminus \mathcal{S}$ that is farthest from all previously selected points in terms of Euclidean distance:

$$\mathbf{p}^* = \arg \max_{\mathbf{p} \in \mathcal{C} \setminus \mathcal{S}} \min_{\mathbf{q} \in \mathcal{S}} \|\mathbf{p} - \mathbf{q}\|_2.$$

This process is repeated until the sampling budget is met. By prioritizing maximal spatial dispersion, this method reduces redundancy and ensures broad coverage of diverse propagation conditions, which is particularly beneficial under extreme sparsity (e.g., 0.02%). The sampling method runs in under 1.6 ms per map and improves RMSE by an additional 0.10 dB on our validation split.

### 3.6. Inference Pipeline

At inference time, each five-channel padded tensor is processed through the network. The output is cropped back to the original field of view and converted to dB scale. The average inference time is 2.5 ms per map, enabling real-time

**Table 1**: Ablation results on our validation set (B21–B25).

| Variant | RMSE |
| --- | --- |
| Baseline (3 ch, no GT) | 4.2401 |
| + 5-channel input | 3.6866 |
| + Phase 2: ReduceLROnPlateau | 3.1285 |
| + Transfer learning (3→1 frequency) | 2.0535 |
| + Data augmentation | **1.5139** |

prediction at the building level—orders of magnitude faster than conventional ray-tracing methods.

## 4. EXPERIMENTS

### 4.1. Dataset and Metrics

We independently split the dataset by assigning buildings B1–B20 for training and B21–B25 for validation, with each raster padded to a resolution of 530×610. The evaluation metric is root-mean-square error (RMSE) in dB after clipping predictions to the range [13, 160]. This clipping reflects the physical constraints of the simulation setup, as the organizers stated that all ground-truth PL values lie within this range. Thus, it ensures that predictions remain within realistic bounds for the specified frequency and indoor layout.

### 4.2. Implementation Details

All models are trained on a single NVIDIA A100 GPU. Batch size is set to 2. Training Phases I and II take 7 hours and 90 minutes respectively. Inference requires 2.5 milliseconds per map.

### 4.3. Ablation Study

Table 1 summarizes the incremental improvements brought by each architectural and training component. All results are based on the 0.5% supervision case and averaged over five validation folds. "no GT" indicates that neither the binary sampling mask $M$ nor the masked ground truth $M \odot G$ is provided; only the physical channels $[R, T, D]$ are used. Notably, using cosine annealing in both Phase I and II leads to degraded performance (RMSE = 4.0641), likely due to overly aggressive learning rate decay during fine-tuning. In contrast, when Phase I uses cosine annealing and Phase II switches to a ReduceLROnPlateau strategy, the model achieves a more stable descent to 3.1285. This suggests that fine-tuning benefits from adaptive learning rate schedules tailored to validation performance. Among all ablations, transfer learning yields the largest single performance gain (-1.075 dB), while data augmentation provides a further significant reduction of -0.5396 dB. Additionally, we performed an ablation comparing models trained with only MSE loss versus MSE com-

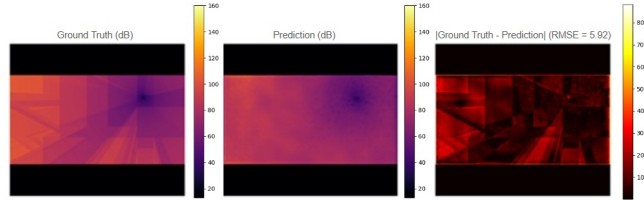

**Fig. 1**: Ground truth (left), predicted PL map (middle) and absolute errors (right) for a representative case. Color bars are included to improve interpretability.

bined with Spread Loss. On the B21–B25 validation split, the combined loss yielded a 0.07 dB RMSE improvement. However, qualitative analysis suggests that Spread Loss may oversmooth behind-wall areas and sharp transitions (e.g., LoS/NLoS boundaries). We also conducted an ablation to evaluate the effect of excluding sampled ground-truth pixels from the loss computation. Including these pixels in the loss increased the RMSE by 0.08 dB on the validation split (B21–B25), confirming that concentrating supervision on unobserved regions enhances interpolation performance under sparse conditions.

### 4.4. Qualitative Results

Figure 1 shows representative examples of poor predictions alongside ground-truth maps and their absolute errors. Error hotspots are typically found in deep-indoor zones or behind walls, indicating the presence of multipath artifacts not well captured by local cues.

Notably, certain blurry regions in the prediction maps fail to replicate sharp discontinuities present in the ground truth. This may be attributed to the smoothing effect of the spread loss, which—while generally effective—can undesirably flatten sharp transitions, potentially degrading performance in such regions.

### 5. DISCUSSION

**Sparse Supervision and Interpolation**

Under extremely sparse conditions (e.g., 0.5%), excluding the sampled points from the loss computation allows the network to focus entirely on interpolating the unknown regions. This implicitly encourages spatial consistency while reducing task complexity. The spread loss supports this by suppressing localized prediction fluctuations and encouraging continuity. However, as shown in our qualitative results, it may also overly smooth abrupt transitions—particularly behind walls—leading to blurry predictions. This suggests a tension between regularization and fidelity in spatial interpolation tasks.

**Effectiveness of Frequency Transfer**

Our two-phase training scheme, consisting of multi-frequency pretraining followed by band-specific fine-tuning, yields the largest performance gain in our ablation study. The initial phase teaches the network generalizable low-level features, while the fine-tuning stage adapts high-level patterns specific to the target band (868 MHz). This approach not only prevents overfitting but also leverages cross-band regularities, resulting in a 1.075 dB RMSE reduction compared to single-band training.

**Limitations and Future Directions**

Task 2 scored only marginally lower than Task 1; this small gap is likely due to statistical variability inherent in our heuristic ESB sampling, which—although it usually aligns points with edges and strata—can still overlook critical propagation hotspots in complex layouts and leave the model under-constrained at those locations. The ESB policy is therefore effective yet inherently heuristic and may miss other important areas in spatially uniform or repetitive environments. Additionally, it assumes pixel-aligned transmitter positions and does not account for angular propagation effects such as beamforming. Future work could explore uncertainty-aware or learning-based sampling strategies that adaptively balance informativeness and coverage, and incorporate physics priors to better capture sharp transitions. We also evaluated ESB sampling under the 0.02% budget and observed a slight 0.002 dB degradation compared to random sampling. While this is likely within statistical variation, it suggests that under extremely sparse settings, the heuristic coverage by ESB may be insufficient to guarantee consistent improvements across all layouts. Future work could incorporate statistical comparisons or ensemble evaluations across multiple seeds to strengthen robustness claims.

### 6. CONCLUSION

We proposed Sparse-Guided RadioUNet, a five-channel U-Net architecture that integrates physics-informed inputs with sparse supervision for path-loss map reconstruction. By explicitly excluding ground-truth pixels from the loss computation, the model focuses on interpolating unknown regions more effectively. We also employed a two-phase training strategy consisting of multi-frequency pretraining followed by fine-tuning on the target band (868 MHz), allowing the network to learn general propagation patterns and adapt to band-specific distributions. In addition, we incorporated a spread loss to promote spatial consistency under sparse supervision, while also observing that it may oversmooth sharp transitions—such as those behind walls—leading to slight blurring in some predictions. With this design, our method enables accurate radio map prediction even under extremely sparse conditions with only 0.5% or 0.02% of ground-truth labels.

Moreover, the inference time per building is just 2.5 milliseconds, which is orders of magnitude faster than traditional ray-tracing approaches. Future work will explore uncertainty-based and reinforcement learning–driven sampling strategies, as well as generative priors using diffusion models, to further improve reconstruction performance under extreme sparsity.

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
