# OpenReview forum: "Sparse-Guided RadioUNet with Adaptive Sampling for the MLSP 2025 Sampling-Assisted Pathloss Radio Map Prediction Data Competition"
_IEEE.org/MLSP/2025_SA_Radio_Map_Prediction_Challenge — SA Radio Map Prediction Challenge at MLSP 2025 Oral_

### Official Review · Reviewer_6AVg · 2025-06-03
**Reviewer Comments on "Sparse-Guided RadioUNet with Adaptive Sampling for the MLSP 2025 Sampling-Assisted Pathloss Radio Map Prediction Data Competition"**

**Rating:** 7
**Confidence:** 4

**Review:**

The paper presents a robust method leveraging a two-stage training strategy to enhance prediction accuracy while achieving ultra-fast inference speed, demonstrating significant potential for real-time systems. To further strengthen the manuscript, the following questions and suggestions are posed for the authors’ consideration:
1. Motivation for U-Net Input Channels: The architecture includes both the binary sampling mask (M) and masked ground truth (M ⊙ G) as input channels. Intuitively, masked ground truth alone might seem sufficient to indicate observed/unknown regions. Are there ablation experiments demonstrating that explicitly incorporating the binary mask (M) consistently improves prediction accuracy compared to using only M ⊙ G? Clarifying the distinct roles of these two inputs would enhance the architectural rationale.
2. Effectiveness of Spread Loss: The introduction of Spread Loss is innovative, but the paper acknowledges that over-reliance on this loss term may over-smooth sharp transitions (e.g., behind walls), leading to blurry predictions. Could the authors provide ablation results comparing models trained with only MSE loss versus the combined MSE + Spread Loss? Quantitative comparisons (e.g., RMSE, structural similarity) would better validate the net benefit of incorporating spatial regularization.
3. Distance Strata Sampling Strategy: In Section 3.5, the distance raster (D) is divided into near/mid/far bins with equal sampling counts per stratum. However, the specific criteria for bin division (e.g., distance thresholds) are not detailed. Additionally, since Section 4.4 highlights larger prediction errors in far-distance regions behind walls, would a non-uniform sampling strategy—allocating more samples to far regions and fewer to near regions—potentially improve accuracy? The authors may consider discussing the rationale for equal sampling or exploring adaptive allocation in future work.
4. Validation of ESB Sampling Effectiveness: The paper states that the proposed ESB sampling improves RMSE by 0.10 dB on the validation set, but if the model’s performance under ESB sampling is marginally lower than random sampling in certain scenarios, this could question the strategy’s robustness. Are there additional experiments (e.g., across different building layouts or sampling rates) that consistently demonstrate ESB’s superiority over random sampling? Further statistical analysis (e.g., t-tests) or visualizations of sample distribution impacts would strengthen the claim.
5. Rationale for Prediction Clipping: In Section 4.1, predictions are clipped to the range [13, 160] dB. Is this clipping based solely on ground truth (GT) value observations, or does it align with physical constraints of pathloss measurements (e.g., minimum/maximum feasible values for the target frequency band)? Providing a physical interpretation for this preprocessing step would enhance the methodological transparency.
Summary: These questions aim to refine the technical justification, experimental rigor, and physical interpretability of the proposed approach. Addressing them would further solidify the paper’s contributions and clarify its practical implications for radio frequency modeling.

---

### Official Review · Reviewer_7yTM · 2025-06-08
**Reviewer's comments on "Sparse-Guided RadioUNet with Adaptive Sampling for the MLSP 2025 Sampling-Assisted Pathloss Radio Map Prediction Data Competition"**

**Rating:** 7
**Confidence:** 4

**Review:**

This paper proposes a five-channel U-Net architecture for indoor path-loss map prediction under extreme sparsity, enhanced with physics-informed features, spread loss, and a heuristic adaptive sampling strategy (ESB). The proposed method is evaluated on the MLSP 2025 challenge, achieving good results in the final leaderboard. An ablation study demonstrate clear gains from each component.

The authors could consider the following aspects for improvement of the manuscript:
- Clarify and briefly define each component (edge, strata, boundary) of the ESB acronym upon first use .
- Figure 1 would benefit from clearer labeling or color bars to highlight error regions more effectively.
- The baseline model (without GT or mask), as briefly noted by the authors, is under-optimized; this should be made more prominent in the discussion to avoid misleading comparisons.

---

### Official Review · Reviewer_pV8R · 2025-06-09
**Reviewer's comments on "Sparse-Guided RadioUNet with Adaptive Sampling for the MLSP 2025 Sampling-Assisted Pathloss Radio Map Prediction Data Competition"**

**Rating:** 5
**Confidence:** 4

**Review:**

1) "building-scale ray-tracing data while leveraging only sparse ground-truth samples" is not very accurate: the reflectance, transmittance and distance from Tx are also available, and are used.
2) "How can a neural network be trained under extremely sparse supervision without collapsing to trivial solutions?", "the Indoor Radio Map Dataset [20] introduces the first large-scale benchmark with ultra-sparse ground truth". The supervision is not sparse, the Indoor Radio Map Dataset used for training includes dense pathloss radio maps.
3) "computed only on unobserved pixels (M = 0)": It should probably be M(j,k) or M_{j,k}, because the unobserved pixels {j,k} should be set to zero, not the whole mask matrix M.
4) "In contrast, IPP-Net...": Probably, "in contrast" is not the right transitional expression here.
5) "480×560 crops with zero-padding" : Why were those values chosen? How does this serve as a data augmentation?
6) "sparse subset of sample points (typically 0.02% or 0.5% ...)" : Why typically? There are only two possible values.
7) The Spread Loss is not mathematically defined.
8) "Inspired by the classical anisotropic Total Variation (TV) regularizer [22], we extend the idea by (i) weighting each gradient by the sampling mask M, and (ii) cosine-annealing the coefficient λ during training. " Here, does i) mean that only the variation at the sampling points are considered in the overall loss function? And for ii), the cosine-annealing is neither defined nor referenced.
9) Overall, I do not see sufficient reasons to introduce a new term (Spread Loss), since it does not seem to be any different than TV loss.
10) Table 1: What does "no GT" mean?
11) "To prevent trivial learning from the sparse ground-truth locations, we exclude sampled pixels from the loss computation.", "By explicitly excluding ground-truth pixels from the loss computation, the model focuses on interpolating unknown regions more effectively.": The authors do not provide any evidence for their claims. Why would this exclusion lead to more effective interpolation? Any numerical experiments?
12) What does "task entropy" mean? The unnecessary and probably inaccurate use of specific terminology should be avoided.
13) It is not clear what the authors mean by "high-frequency noise".